## PERSPECTIVE

# A model for person perception from familiar and unfamiliar voices

Nadine Lavan [1]✉ & Carolyn McGettigan[2]✉

When hearing a voice, listeners can form a detailed impression of the person behind the voice. Existing models of voice processing focus primarily on one aspect of person perception - identity recognition from familiar voices - but do not account for the perception of other person characteristics (e.g., sex, age, personality traits). Here, we present a broader perspective, proposing that listeners have a common perceptual goal of perceiving who they are hearing, whether the voice is familiar or unfamiliar. We outline and discuss a model - the Person Perception from Voices (PPV) model - that achieves this goal via a common mechanism of recognising a familiar person, persona, or set of speaker characteristics. Our PPV model aims to provide a more comprehensive account of how listeners perceive the person they are listening to, using an approach that incorporates and builds on aspects of the hierarchical frameworks and prototype-based mechanisms proposed within existing models of voice identity recognition.

When we hear a voice, we instantly form an impression of the person who is talking: This process can be referred to as person perception from voices. Person perception from voices can include the processing of any number of person characteristics: Is the person we are hearing likely to be male or female? Are they young or old? Do they sound like they are friendly, aggressive, or shy? Are they talking in their first language? If we know the specific person who is speaking, we may also be able to identify and name them.

Given that listeners may perceive a wealth of characteristics about the person they are hearing, it is surprising that existing cognitive and neural models of voice processing primarily focus on recognising the unique identity of a familiar person[1,2]. Within these models, the perception of person characteristics from unfamiliar voices is considered only in terms of 'becoming familiar' with a unique identity (e.g. ref. [3]) or via listeners being able to discriminate between two voice identities (e.g. ref. [4]). Indeed, most of the extant theoretical literature considers familiar and unfamiliar voice (identity) processing as being both qualitatively and mechanistically distinct processes[4–7].

However, this framing of unfamiliar voice (identity) perception seems to be at odds with our subjective experience of familiar and unfamiliar voices alike as being rich in cues to person characteristics[8]. In this paper, we therefore put forward a new and more comprehensive account of person perception from both familiar and unfamiliar voices. We will first briefly review existing models of voice identity perception and discuss how familiar identity perception may be a special case of person perception. We will then outline the evidence for the wide range of person characteristics that can be perceived from (unfamiliar) voices, beyond person identity. We suggest that outside of experimental contexts, person perception from unfamiliar and familiar voices alike has a common goal of recognising who is being heard, but with crucially many more possible perceptual outcomes than recognising a unique individual. To achieve this common goal, we argue that instead of having distinct mechanisms for familiar vs unfamiliar identity perception, person perception from all voices employs a common mechanism involving

[1]Department of Experimental and Biological Psychology, Queen Mary University of London, London, UK. [2]Department of Speech, Hearing, and Phonetic Sciences, University College London, London, UK. ✉email: n.lavan@qmul.ac.uk; c.mcgettigan@ucl.ac.uk

the recognition of different person characteristics, be they identity-specific (for familiar voices) or not.

## Existing models of voice (identity) perception

Over the last two decades, several models of voice (identity) processing have been proposed. These models were created based on evidence from different literatures and, as a result, often emphasise distinct aspects of voice processing. In the following section, we will outline the most prominent models, sketch out their remit, and show where these models differ from and/or complement each other.

**A hierarchical model of (familiar) voice processing.** The argu-ably best-known and most general model of voice processing has been proposed by Belin and colleagues[1,9,10] (see Fig. 1), which has, in turn, inspired more recent and updated expansions of this model[3,11]. This model takes Bruce and Young's model of face perception[12] and directly applies it to voices: Following a low-level auditory analysis and a structural analysis of vocal sounds, pro-cessing continues along three functionally independent but partially interacting pathways. These pathways support the perception of speech, emotion, and identity information, respectively. Identity recognition is achieved in a 'voice recognition unit', which is 'acti-vated by one of the voices known to the person' (p.131[9]). Voice recognition units interact with face recognition units (not shown in Fig. 1) and once recognition in one or more modalities is achieved, a putative amodal 'person identity node' is activated.

**A mechanistic account of familiar voice identity recognition.** Prototype-based coding is at the heart of a model of (familiar) voice identity perception by Lavner and colleagues[2], which pro-vides a mechanistic account of familiar voice recognition. In this model, familiar voices are thought to be represented in terms of their (acoustic) deviation from a voice prototype. This prototype is described as being an average of all the voices that have been encountered by a listener. When a voice is heard, its deviant features in relation to the prototype are computed and the result of this deviant feature extraction is then matched to existing

reference patterns or representations of known voices. If the distance between the deviant features of the perceived voice and the reference pattern is sufficiently small, the voice will be recognised as belonging to a particular familiar person. For a visualisation of the processing steps, see the final figure in this paper, which has been adapted from Lavner and colleagues' ori-ginal model[2]. How person perception of unfamiliar voices may work is not specified within this model[2]: Although the model includes a loop to indicate that perceived voices that do not fit an initial reference pattern are iteratively compared to other refer-ence patterns— presumably until the matching voice is found and the familiar voice is recognised—there is no proposed mechanism for how truly unfamiliar voices are processed or are at least recognised as being unfamiliar.

**Combining hierarchical and mechanistic models.** While Belin and colleagues' hierarchical model of voice processing[9,10] outlines a pathway to voice identity recognition, it does not specify the mechanism by which identity recognition is achieved within its proposed processing hierarchy. However, the framework can readily incorporate the mechanistic model proposed by Lavner and colleagues[2]. These two models have indeed recently been combined by Maguinness and colleagues[3], who further extended this combined model to propose a mechanism for how unfamiliar voices become familiar, thus modelling voice identity learning. Maguinness and colleagues[3] propose a hierarchical framework for voice identity perception, in which deviant features are extracted in relation to a prototype upon hearing familiar and unfamiliar voices alike. The model includes very similar hierarchical processing stages to Belin and colleagues' model[9,10] and incorporates a prototype-based recognition mechanism as described by Lavner and colleagues[2]. To model voice identity learning, Maguinness and colleagues[3] propose that at first exposure to an unfamiliar voice, its pattern of deviant features does not fit any reference pattern for familiar voices and, therefore cannot be recognised. Over repeated exposures, however, the pattern of deviant features will eventually become an established stored reference pattern or representation and can thus become recognisable as a familiar voice.

**Distinguishing between familiar voice recognition and unfami-liar voice discrimination.** Kreiman and Sidtis[5] present a view of voice identity perception that is strongly influenced by evidence from neuropsychological research. The authors propose that unfamiliar and familiar voice identity perception are two mechanistically distinct processes. They cite neuropsychological studies, in which individuals with brain injuries completed two types of identity perception tasks: (1) Identity recognition, which traditionally requires the listener to link a single presented voice to a specific known identity and (2) identity discrimination, in which the listener compares two voice samples and determines whether they were made by the same person. These studies showed a double dissociation, where individuals who had difficulties in voice identity processing (known as phonagnosia) following a brain injury were able to successfully complete identity discrimination tasks with unfamiliar voices but could not recognise familiar voices, and vice versa[4,13,14] (see similar reports of dissociations in developmental phonagnosia[15–17]). This evidence is compelling and conclusively demonstrates that discrimination is not a prerequisite for recognition[4]. However, as other authors have noted[3,18], manip-ulations of voice familiarity and experimental task overlap in these studies. Thus, although discrimination tasks were crucial in allowing neuropsychologists to characterise identity perception impairments in phonagnosia, the conflation of task with voice familiarity makes it unclear whether the dissociations reflect

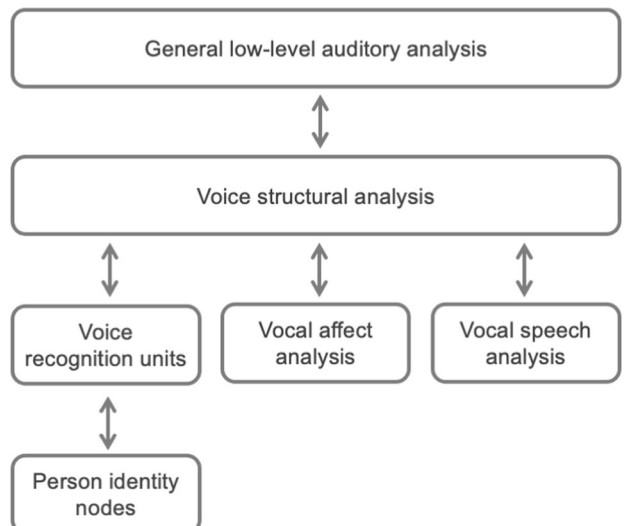

**Fig. 1 A hierarchical model of voice processing redrawn and adapted from Belin et al.[9].** This model proposes that voice perception is a series of hierarchically organised processes, starting from the low-level analysis of the sound, an analysis of the "structure" of the voice, followed by three separate (but partially interacting) pathways for identity, affect, and speech processing.

fundamental differences in familiar versus unfamiliar voice identity processing per se, differences in task demands, or aspects of both. Although the original studies are careful to discuss their results in light of both the experimental task and voice familiarity, later research has often assumed that familiar and unfamiliar voice processing are entirely distinct processes without explicitly considering task demands (see ref. [7] for a recent review in support of this assumption).

**The scope and remit of existing models**. What emerges from this overview is that existing theoretical models focus on identity perception, specifically familiar voice identity recognition[2,9]. Where the perception of unfamiliar voices is considered[3,4], it is likewise always in the context of voice identity perception, proposing, either implicitly or explicitly, that familiar vs unfamiliar voice (identity) perception are distinct from one another. The models thus only consider one example of when and how listeners are trying to make sense of which specific person they are talking to (i.e. identity-specific perception) and neglect to account for what listeners may do when identity-specific recognition is not possible (or not the focus of attention). That is, even though the perception of other person characteristics from (unfamiliar) voices, such as impressions of age, sex, regional accent, or perceived personality, is as much part of person perception as is identity-specific recognition, these core aspects of person perception are currently missing from theoretical models (though see Bruce and Young[12], who describe 'visually derived semantic codes' for unfamiliar face perception).

## Recognising familiar voices is a special case of person perception from voices

The perception of familiar voice identities has captured the imagination of researchers. Identity-specific perception is forefronted in the voice perception literature, with the recognition and perception of (personally) familiar voices further being viewed as being 'special', whereas unfamiliar voice identities are thought to hold no such special status and to lack (social) importance in perception[5,6]. Below, we will briefly review some themes emerging from the (familiar) voice identity perception literature.

Familiar voices are the voices of people whose identities are known to us, for example, because they are famous (e.g. actors and politicians), known to us in our personal lives (e.g. family and friends) or because we have been introduced to them as part of an experimental paradigm (e.g. recognition training, passive exposure). Studies have shown that listeners can recognise familiar others from their voices, although the accuracy of identity recognition depends on the selection of voices/stimuli being heard and the characteristics of the listeners, as well as other factors[5,19]: For example, longer voice recordings have been shown to increase identification accuracy for famous voices[20,21], while acoustic manipulations (e.g. filtering or shifting of acoustic cues) or intentional voice disguise generally reduce the accuracy of familiar voice identity recognition[22–26]. Further, tasks including within-person variability (e.g. a person speaking vs shouting) pose perceptual challenges to familiar (and unfamiliar) voice identity perception (e.g. refs. [18,27]). The type and degree of familiarity with a voice also matter: Personally-familiar voices (e.g. romantic partners) are recognised with very high accuracy even in perceptually-challenging tasks, while accuracy for lab-trained voices is substantially lower within those same tasks[26,28].

Identity-specific recognition is peculiar to familiar voices: Only a familiar person can be uniquely recognised—by name or otherwise—based on their voice, as 'recognition' requires some stored representation of the specific thing that is to be recognised. This is not possible for unfamiliar voices, as a stored person-specific representation cannot exist for someone that we have never encountered before. This possibly sheds light on how the literature has come to see familiar and unfamiliar voice identity perception as distinct: It is impossible to run an identity recognition task with previously unheard voices—there is little point in asking a listener whether an unfamiliar voice is Tom or Dave—thus often necessitating the use of an alternative experimental task, such as voice identity discrimination. However, while this is a practical solution, we argue that specific identity perception is not a priority when hearing unfamiliar voices in everyday life. For example, we can easily follow spoken conversations in terms of who is talking and when, without any need to perceive the specific voice identities of the speakers. In our view, explicit voice identity discrimination tasks may be valid in experimental contexts, but they bear little resemblance to how person-related information is processed from unfamiliar voices in everyday life.

Another unique aspect arising from familiar voice identity perception is that (personally) familiar voices, at first glance, appear to be much richer, personally-relevant signals than unfamiliar voices[5,6,29]. In addition to the representation of the sound of a familiar voice, specific memories associated with the person will be accessible to a listener once a voice is recognised. The sound of a familiar voice may also activate representations of the person's face[30,31], alongside biographical knowledge about them (even for people we 'know' but have never met, such as celebrities). Similarly, we can access emotionally and socially salient information about, for example, whether we like this person or not, as well as specific memories of events and situations involving them[5,6]. The emotional salience that may accompany hearing a familiar voice is readily expressed in the statement 'it's so good to hear your voice'; and by—albeit anecdotal—accounts of emotional responses to voice recordings of personal significance (e.g. the story of a widow's fondness for listening to the London Underground announcements recorded by her late husband[29]). However, unfamiliar voices can still be emotionally evocative as a result of salient personal memories: For example, we may have an emotional reaction to hearing someone speak in an accent that we have not heard since childhood, or hearing someone who sounds like a familiar individual. These situations are, however, likely to be exceptions, and not the rule, for unfamiliar voice perception.

Generally speaking, the experience of recognising a familiar voice is therefore more likely to trigger experiences that are more personally relevant, specific, and richer compared to the experience of an unfamiliar voice. Thus, there are clearly aspects that are unique to familiar voice processing compared to the perception of unfamiliar voices. However, while these unique aspects can perhaps go some way to explaining the literature's focus on identity perception from voices, this does not mean that person perception from unfamiliar voices is uninteresting and can be neglected—after all, even the most treasured and familiar voices were once entirely unfamiliar to us.

## Complex impressions of a person are formed from unfamiliar voices

If a voice is unfamiliar, what then is left for a listener to perceive about its owner in the absence of identity recognition? In this section, we will outline evidence showing that being unfamiliar with a specific talker in no way precludes person perception—in fact, listeners can and do routinely form rich and complex impressions of a person without being familiar with them or knowing their unique identity (see Fig. 2).

Empirical evidence from different fields, including evolutionary and social psychology, sociophonetics and bioacoustics, demonstrates that human listeners will readily provide judgements of many

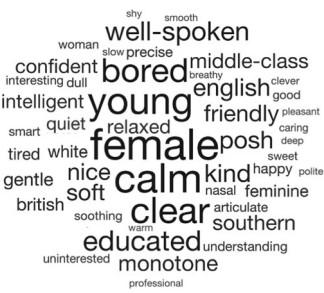

**Fig. 2 Word cloud illustrating the multivariate nature of person perception from unfamiliar voices.** This word cloud shows descriptions of people that listeners provided of six unfamiliar voices (see Lavan[8]). The talkers were all female, speaking with a Standard Southern British accent and were heard reading out linguistically-neutral sentences for 2 min. Listeners noted down words and short phrases that described the people they heard. Font size scales with the frequency with which individual descriptors were mentioned.

person characteristics from an unfamiliar voice. Listeners perceive physical characteristics, such as the sex of a person[32–34], the person's age[32–35], aspects of body size such as height and weight[36,37] and physical strength[38–40]. Listeners furthermore readily make judgements about other (perceived) social or psychological characteristics based on hearing a voice. These include a person's spoken accent[41], native speaker status[42], sexual orientation[43,44], social status[45] and occupation[46]. A recent focus of interest has been on 'first impressions' of personality traits (e.g. trustworthiness and competence). This work has, to some extent, shadowed prior research on the face and social category perception[47,48]. Research obtaining evaluations of speaker traits from voices— typically via rating scales—has found that first impressions are formed rapidly[49–51], are typically made with high inter-rater reliability for most trait categories, can be mapped onto a small number of underlying dimensions (e.g. warmth/trustworthiness vs dominance[49,52]), and are consistent across different types of vocal stimuli (e.g. different languages and linguistic content[53–55]). While there is overall little empirical evidence that first impressions of a person are linked to their actual personality or character traits[56], these subjective impressions can nonetheless influence our daily interactions, behaviours and decisions (e.g. refs. [57,58]).

The studies listed above examine the perception of a wide range of person characteristics. Yet, there is considerable heterogeneity in both the theoretical motivations and the methodological approaches across disciplines. For example, while researchers in bioacoustics are often concerned with how actual physical variation in bodies gives rise to differences in voice acoustics, and evolutionary psychologists are interested in how these might signal characteristics such as physical or reproductive fitness to potential adversaries or mates, social and experimental psychologists have been more focused in how perceived personality traits in the voice might relate to each other and a range of social outcomes for the speaker (e.g. employment, election, financial investment). These differences in research questions mean that different types of person characteristics—for example, height versus trustworthiness—are often studied completely separately, using different tasks and judgements (e.g. various types of scales vs categorical judgements vs judgements relative to the listener's own characteristics), and with varying focus on the accuracy of perception and its relationship to a 'ground truth'. So, although this body of work illustrates the potential richness of person percepts from voices, it is difficult to gain a full appreciation of how listeners form (complex) impressions of unfamiliar others from voices based on any individual study or field. Similarly, it remains difficult to gauge the relative salience and

importance of different perceived characteristics in relation to one another.

Recent studies in experimental psychology have taken a broader view on person perception from (unfamiliar) voices, attempting to examine which characteristics are spontaneously perceived from unfamiliar voices, and how multiple person percepts might be organised in relation to one another. Lavan[8] asked participants to listen to a recording of a voice and provide a list of words that described the person they heard, thus allowing the listeners themselves to generate the characteristics rather than making judgements on characteristics predetermined by the experimenter (see also Pear[46]). Listeners provided a wide range of descriptions based on the voices, including descriptions of physical, psychological, and social characteristics (see Fig. 2 for a word cloud showing an overview of these free descriptions), showing that listeners do form complex impressions of other people based on their voices. Additionally, the study reports that there was a structure to listeners' responses—listeners tended to first mention physical characteristics, like sex and age. Further recent evidence for such structured person perception has come from perceptual gating-type studies, in which listeners were exposed to brief voice clips of increasing exposure duration (e.g. from 25 to 800 ms) and asked at each duration to provide ratings of a range of physical, social and psychological characteristics. High inter-rater agreement at any exposure duration was interpreted as an impression having been formed. These results also indicated a temporal hierarchy of person percepts, where impressions of physical characteristics (age and sex) are formed more quickly than impressions of social and trait characteristics[50,51] (see also ref. [59] for faces).

The review above clearly shows that listeners can perceive a wealth of person-related information from unfamiliar voices. Indeed, the main impression listeners take away from encountering an unfamiliar voice is unlikely to be limited to identity-specific information (e.g. 'I did not know this person'; 'Person X sounds different from Person Y in this conversation'), as current models might perhaps suggest. What is much more likely is that the listener comes away with a (possibly hierarchically organised) combination of first impressions covering aspects of physicality, social characteristics, and other traits (e.g. 'This young woman sounded really friendly and clever'). Taken together, the findings from unfamiliar voice perception thus strongly motivate the need for existing models of voice perception to be updated to include person perception beyond identity recognition.

## A model for Person Perception from Voices (PPV)

Models of voice perception need to account for the goal of person perception from familiar and unfamiliar voices alike, instead of framing the success or failure of the perception of specific identities as the primary, and perhaps only, objective. Outside of experimental tasks, listeners' shared aim is to perceive who a person is at a holistic level, which is not restricted to identity-specific perception, but is instead based on whatever information about a person is available and meaningful to them. We propose that person perception from familiar and unfamiliar voices alike can be achieved via a common mechanism of recognition, which includes the recognition of specific known identities, but also any number of person-general characteristics (such as age, sex, etc.).

As it stands, the voice identity perception literature mainly operationalises 'recognition' as a process during which a voice is attributed to a specific person, thus primarily considering recognition in relation to familiar voices and identity-specific perception. When listeners recognise a familiar person, they can find themselves being able to name the person ('It's Aisha') or identify them in the absence of naming ('It's the actor from those superhero movies'). In contrast, (accurate) identity recognition

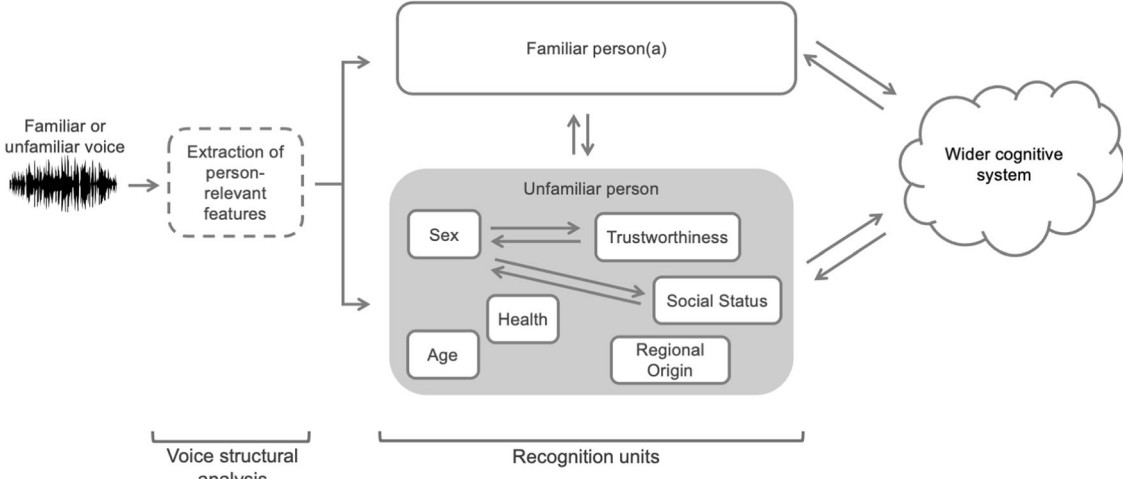

**Fig. 3 A hierarchical model of person perception from voices (PPV).** The model illustrates how different person characteristics may be recognised and may interact and inform each other during person perception from voices. Boxes with a solid outline describe a recognised percept and the box with a dashed outline describes a computational process. Brackets at the bottom of the figure loosely map the processing stages of the PPV to Belin et al.'s hierarchical model of voice identity perception[10]. Not all possible interactions between percepts are shown.

from an unfamiliar voice is impossible. However, upon hearing an unfamiliar voice that is the fairly high-pitched, likely youthful and likely female with a Californian accent and a lot of vocal fry, listeners can readily recognise that they are listening to a young woman as opposed to a child. Depending on prior experience, listeners may even recognise that the voice described above fits the persona or social stereotype of 'Valley Girl'[60]. The impression of an unfamiliar person from a voice—be that based on a few broad demographic characteristics or a well-formed persona— provides meaningful information about that person and arises in the absence of listeners being familiar with the person's unique individual identity. Crucially, this impression can be entirely subjective and does not have to be accurate to exist and to affect behaviour. Our proposal also eliminates the need for two mechanistically entirely distinct processes for perceiving information about people and their specific identity from familiar vs. unfamiliar voices (e.g. recognition vs discrimination[4,5,7]).

Here, we therefore outline a model for person perception from voices (PPV, Fig. 3), which describes how listeners may recognise one or more aspects of a person or a persona to provide a more comprehensive account of listening situations that result in person perception from vocal cues. The PPV model includes a possible route to account for the recognition of familiar persons or personae (see below) and adds to this the capacity for recognition of one or more individual person characteristics (sex, age, health, etc.). Although visually depicted as two routes, these processes are not mutually exclusive to one another: While person-specific recognition may be prioritised depending on the listening situation, person-general characteristics may be recognised in parallel with recognising a familiar person/persona.

We first consider a direct route for the identification of a familiar person or persona. This part of the model is closely aligned with previous theoretical accounts of identity perception where, following the extraction of person-relevant features from a heard voice, this voice is recognised in a person-specific 'recognition unit' as a known person without any intermediate processing steps (see refs. [3,5,9,11]). In our model, we extend this mechanism beyond identity-specific recognition to additionally support the recognition of known personae. Personae include familiar social stereotypes, such as a Valley Girl or City Banker, where the voice yields a coherent and detailed percept of a known type of person without the identification of a specific individual.

We next consider a situation in which listeners hear an unfamiliar voice, from which a specific person or a persona cannot be recognised. When perceiving an unfamiliar voice, listeners recognise one or more physical, psychological, and/or social attributes of a voice. As indicated by the grey box in Fig. 3, the recognition of several person characteristics may give rise to a complex percept of an unfamiliar person comprising the recognised characteristics (e.g. middle-aged, tall, and confident). However, even when the listening situation makes only one or two characteristics available to perception—for example, due to brevity or poor signal quality—the PPV model still allows for these to be recognised as individual percepts.

When multiple person characteristics are recognised, this may happen simultaneously or closely staggered in time, modelling our experience of deriving complex, multi-attribute impressions from unfamiliar voices. Emerging evidence from voice research suggests that this perception of different person characteristics from unfamiliar voices may indeed be structured. For example, impressions of physical characteristics, such as age, sex, and health, are formed more quickly than trait or social characteristics, such as level of education and trustworthiness[50]. Based on this evidence, we tentatively place characteristics like sex and age 'earlier' in the processing chain compared to traits and social percepts in Fig. 3. We furthermore propose possible interactions between characteristics, which can reflect existing patterns of co-occurrence in the population (e.g., sex and height recognition: female talkers are usually smaller than male talkers). Therefore, our model allows for (some) bidirectional effects of facilitation (or inhibition) of any of these characteristics on one other (e.g. a percept of 'female' may facilitate a percept of 'smaller than average height'). Further investigation will need to test whether such interactions of representations primarily happen within the recognition process, for example, through direct mutual co-activation of stored representations, or via yet another route implicating additional systems (e.g. top-down effects of attention or memory).

Our model also includes the potential for bidirectional interactions between the recognition of a person or persona and the recognition of person characteristics. The potential for the recognition of person characteristics to influence the recognition of a known person or persona (upward arrow toward 'Familiar Person(a)' in Fig. 3) describes situations in which an objectively

---

**Box 1 ▌ Structuring a perceptual hierarchy: bottom-up, top-down and post-perceptual influences**

The PPV model proposes that multiple person-related characteristics can be recognised. The multi-attribute nature of the model warrants some further consideration of how multiple perceptual representations might be organised and prioritised in perception, both via bottom-up, top-down, and post-perceptual processes.

When and how different characteristics are recognised could be associated with the following bottom-up processes:

- The accessibility of the acoustic cues in time (e.g. pitch cues can be perceived within a few glottal cycles[62], predicting fast recognition of characteristics associated with pitch, such as sex, age, and dominance).
- The relative discriminability of characteristics based on their distinguishing acoustic cues (e.g. pitch is a highly salient cue to differences between adult male and female voices, while it is less reliable for recognising the sex of child voices[72,73]).

Thus, the recognition of certain person characteristics may be achieved more quickly than others that are less clearly marked (e.g. sex perception from adult voices > sex perception from child voices[72]). However, does the bottom-up availability of certain cues make recognition of some characteristics obligatory en route to less accessible or more specific percepts? For example, must a listener recognise that a voice is relatively high-pitched, and therefore likely female, in order to recognise that it belongs to their sister?

Top-down effects could also impact recognition:

- Task demands may require the evaluation of a personality trait, which could down-weight some characteristics in favour of others (participants do not reliably recognise their own voice within a larger set of unfamiliar voices when being asked to make attractiveness judgements[74]).
- General priors and context-specific expectations may affect which representations are prioritised (e.g. failing to recognise a familiar work colleague's voice when heard unexpectedly in a holiday setting).

Finally, higher-order, post-perceptual effects may also influence or override the outcomes of the recognition process. For example, bias for associating lower voice pitch with larger body size mediates the apparent relationship between pitch and rated dominance for male voices, suggesting that listeners use perceived body size as a heuristic for judgements of dominance[75].

Thus, while a basic structure of person perception from voices may exist, determined by bottom-up availability and accessibility of acoustic cues, there must be scope within a processing hierarchy to dynamically re-structure the processing chain in response to highly variable listening situations and perceptual priorities.

---

familiar voice is not immediately recognised as such, but where recognition of the specific identity emerges after additional information about the person has been accumulated from the voice signal in a largely bottom-up fashion (e.g. when hearing a low/moderate-familiarity voice). On the other hand, the potential effects of a recognised familiar person or persona on the recognition of person characteristics (downward arrow from Familiar Person(a) in Fig. 3) are perhaps less intuitive. These can, however, account for previous reports that familiarity with a specific person might affect the neural decoding of person sex and age from their face/voice[59], and evidence that familiarity yields faster judgements of sex from voices[61].

It is possible that an 'indirect route' to (familiar) person or persona recognition via the prior recognition of other person characteristics may, in fact, be the typical means of recognising a familiar person or persona. We highlighted above that there may be a structure or hierarchy to how different person characteristics are recognised: By extension, it may also be the case that, for example, sex recognition is faster than identity-specific recognition[50,59,62], such that listeners may routinely recognise some person-general characteristics before recognising the specific identity or persona. Despite the possible interactions, the PPV model still allows for the direct person/persona recognition via matching to a single stored identity representation as outlined above (adapted from Belin and colleagues[9] and Maguinness and colleagues[3]), where prior or parallel processing of other person characteristics is not obligatory. Future work will therefore need to determine to what extent such a hierarchy may be obligatory, what factors determine how specific people or personae are recognised, as well as what the possible functional benefits of an indirect route to identity recognition may be (see Box 1 for further considerations regarding the structure of a perceptual hierarchy).

Finally, at the top of the PPV hierarchy is the 'wider cognitive system', indicated by the cloud in Fig. 3. We borrow this component from Bruce and Young's model of face processing (in which the authors note: "The 'cognitive system', by convention, is somewhat cloudy.", p. 311[12]). We model communication from the person/

persona recognition machinery to the wider cognitive system, to support the storage and/or retrieval of person-relevant information such as biographical knowledge about a person, episodic memories of social interactions with them, and emotional reactions to them (see also Kreiman and Sidtis' discussion of wider systems engaged when listening to familiar voices[5]). We furthermore allow for effects of the wider cognitive system upon the recognition of person characteristics. These could, for example, include top-down influences of expectations, contextual effects, or attentional modulations on person perception (see Box 1).

## A common recognition mechanism for person perception from voices

To date, there are to our knowledge no mechanistic proposals for how characteristics of a person, other than a specific identity, are recognised from their voice. There are various ways in which a common recognition mechanism—for recognising a person, persona and/or any person characteristic—could be structured. Figure 4 illustrates two possible versions of a common mechanism: the first alternative (Fig. 4a) is built on previous proposals from the voice (and face) perception literature that were heavily inspired by prototype-based accounts (as opposed to episodic-/exemplar-based accounts[63–66]), while the second (Fig. 4b) assumes no involvement of prototypes.

In the PPV model, voices are first encoded and perceptually analysed based on their acoustic features (see also 'voice structural analysis'[10]). A common mechanism might then see voices being compared to a stored voice prototype, which is based on an average(d) voice derived from a person's accumulated experience with human voices (Fig. 4a). The outcome of this process is a pattern of deviant features, which describes how the heard voice differs from the voice prototype. This pattern of deviant features is then compared to stored reference patterns (which are functionally equivalent to 'mental representations'): According to Lavner and colleagues[2] and Maguiness and colleagues[3], these stored reference patterns (representations) also consist of deviant

## a) Recognition mechanism based on a prototype model of person perception (adapted from Lavner et al. 2001)

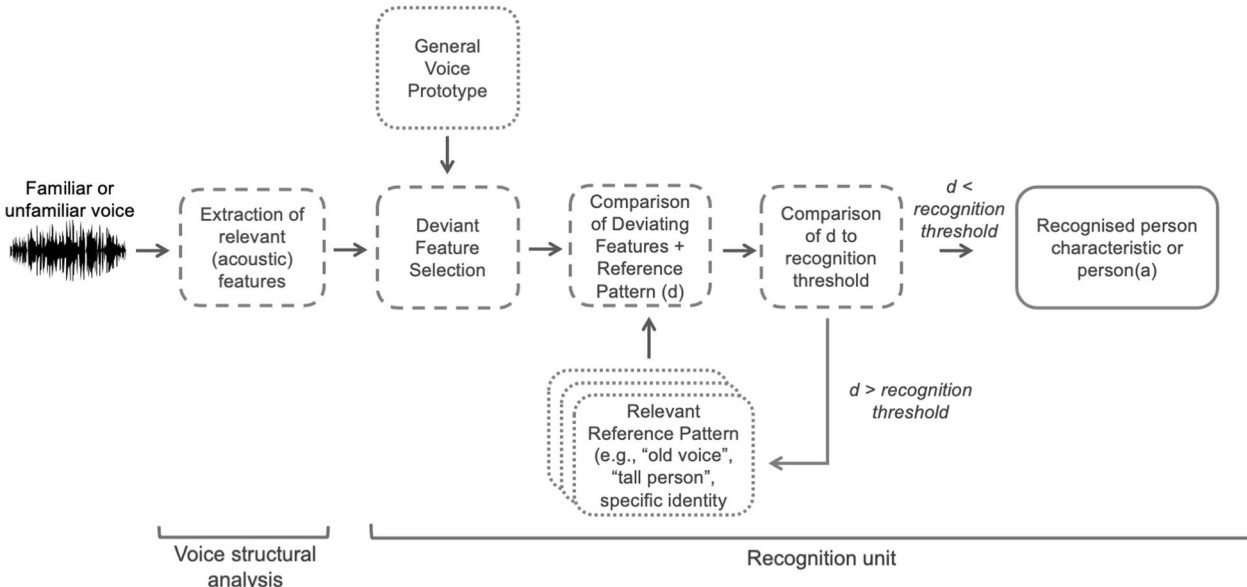

## b) Alternative recognition mechanisms without prototype-based comparisons

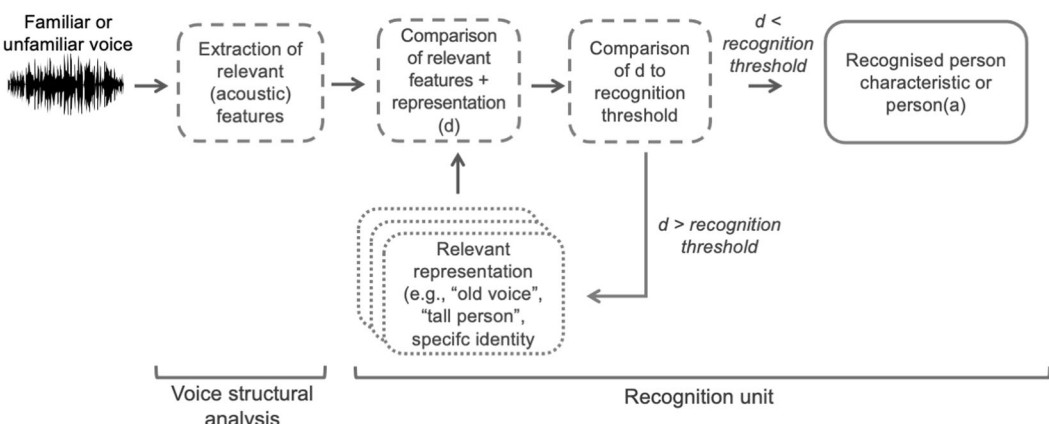

**Fig. 4 Illustration of a common mechanism for person perception from voices.** Two alternative mechanisms for how recognition of a person, persona, or person characteristic may be mechanistically achieved. **a** A mechanistic account of the common recognition mechanism adapted from Lavner and colleagues[2]. **b** An alternative, simplified account of recognition without compulsory prototype-based coding of deviance. Across panels, **a** and **b**, each box with a solid outline describes a recognised percept, each box with a dashed outline describes a computational process, and each box with a dotted line describes a type of mental representation supporting recognition. The box 'recognised person characteristic or person(a)' directly maps onto each of the boxes labelled as percepts (solid outline) in Fig. 3.

feature patterns of how known identities differ from the voice prototype. While previous prototype models of voice identity recognition only consider comparisons to stored identity-specific reference patterns (representations) in order to model familiar voice identity recognition, we propose that the same mechanism may be applied to stored reference patterns for specific known persons, personae, and/or person characteristics (e.g. physical, social, and psychological, see Fig. 4a). In this comparison process, a distance (d) between the deviant feature pattern (of a heard voice) and a stored reference pattern (representing a person, persona or a person characteristic) is computed. If this distance (d) is small enough and thus falls within a recognition threshold, recognition of that person, persona or person characteristic takes place. If the distance (d) exceeds the threshold, the pattern of deviant features for the heard voice is iteratively compared to other relevant stored reference patterns (e.g. another identity or another sex/gender).

While the extant prototype-based model of voice identity perception by Lavner and colleagues[2] may be readily extended to account for person perception more broadly, Fig. 4a only depicts the simplest possible instantiation of a prototype-based mechanism, based on comparisons of incoming signals with a *single* voice prototype. Existing prototype-based accounts for identity recognition propose variations of the model illustrated above where there could be multiple, functionally-relevant voice prototypes (e.g. sex-specific prototypes[3,66]). For a model to incorporate multiple prototypes, we would assume that each prototype would code for an average of a characteristic (identity, age, height, etc.), with a relevant prototype being selected ad-hoc depending on the listening situation. We note, however, that the assumption of multiple prototypes brings with it a substantial increase of complexity within the mechanism. Similarly, there are questions about how, and perhaps more importantly, why any prototype would qualitatively and/or functionally differ from any

---

**Box 2 | Is there a functional role for prototypes in models of voice recognition?**

Prototype-based models of voice (and face) identity recognition were originally invoked to account for empirical findings showing that highly distinctive voices and faces are recognised more quickly and with higher accuracy[2,76].

By basing identity recognition on deviant features relative to a prototype, the magnitude or uniqueness of the deviance pattern increases with increasing distinctiveness. This is used to explain better identity recognition performance for more deviant identities, thus ushering in a functional role for prototypes during identity recognition.

When considering the recognition of other person characteristics, however, a voice that is very deviant from a prototypical "female voice" would likely be less well recognised as a woman (but instead might be recognised as a child or an adult of another gender). Distance from a prototype per se, therefore, does not intuitively account for better recognition of any characteristic (or category) beyond, perhaps, a unique identity.

There are thus two possibilities:

- Identity-specific recognition could truly be mechanistically different from the perception of other types of characteristics, warranting the existence of a prototype that is functionally distinct from a representation and that shapes the nature of the representation.
- Prototype-based coding of deviance may be functionally not obligatory for either identity or person perception (see Fig. 4b), and empirical findings underpinning prototype models for identity recognition can be explained otherwise.

Should there be a—perhaps limited—functional role for prototype-based coding, prototypes and representations are traditionally proposed to be functionally and qualitatively distinct: The prototype is an average voice, while representations are patterns of deviance in relation to this prototype. When outlining mechanisms of identity recognition, some authors also mention the use of sex-specific female and male prototypes—others even propose 'multiple' prototypes the nature of which remains otherwise unspecified[2,3,77].

As a result, there are open questions with regard to how a model with multiple prototypes may function:

- How many prototypes are there, or should there be?
- In what order would different prototypes be engaged in calculating deviance patterns (e.g. does speaker sex need to be recognised first, in order to select the 'female voice' prototype to support recognition of other characteristics)?
- What would the outputs of multiple prototype comparisons be (e.g. sequential prototype comparisons creating deviances of deviances vs. individual deviance patterns for all invoked prototypes)?

In models with obligatory prototype-based coding, the answer to each of these questions would affect even the most fundamental aspects of a recognition mechanism. Given that there are questions about whether prototype-based coding of deviance is required in the first place[65,78] and what the nature and number of functionally-relevant prototypes may be, it is worth considering whether recognition could be achieved without prototypes. Please see Fig. 4b for such a simplified mechanism.

---

representation. This latter issue is exacerbated when considering multiple, potentially co-dependent, recognition processes as the PPV model does, therefore requiring a large number of prototypes. We expand on these issues in Box 2.

Given the conceptual complexities that arise from obligatory prototype-based coding (be there one or multiple prototypes), we propose an alternative version of a common mechanism for recognising a familiar person, persona and/or different person characteristics (Fig. 4b). Specifically, the mechanism in Fig. 4b removes voice prototypes as a functional component from the recognition process, such that the extracted person-relevant acoustic features of perceived voices can be directly compared to relevant stored representations. This alternative mechanism thus fully preserves one of the core ideas behind the original recognition mechanism (matching of relevant features of a heard voice to stored representations) but removes the original two-step mechanism introduced by prototype-based coding (deviant feature selection + comparison of deviant features to stored representation). By removing the functional role of a prototype, the distinction between (average-based) prototypes and (deviance-based) representations becomes obsolete. Further, in the absence of a functionally-relevant prototype, the content of stored representations must change from being stored patterns of deviance (in relation to a prototype) to retaining and encoding all the person-relevant features. Whether this simplified approach is more reflective of the cognitive processes for perceiving people from voices than the prototype-based model remains to be tested.

We stress at this point that we do not wish to claim that prototypes (or at least representations of what a central tendency of a category would be) do not exist: Listeners can have notions of what a prototypical human voice (vs. non-human voices) and a prototypical female voice (vs a male voice) sound like. They may even have a notion of what sounds prototypical for a specific voice identity (vs another identity). Prototypes may therefore structure our experience of different types of voices. However, like some previous work[67], we raise questions about whether prototypes should (1) be included as functionally-obligatory components of recognition mechanisms, and (2) whether this inclusion should, as a result, be able to define the nature and content of representations by mandating purely deviance-based encoding and storage of information for representations[2,3].

Beyond what the different functional components of a common recognition mechanism may be, an expanded model of person perception requires some additional considerations about the nature and content of representations (e.g. categorical vs continuous information; relationships between supra- vs. subordinate category members). We discuss these in Box 3.

## Outlook

The PPV model aims to provide a more comprehensive account of how listeners make sense of the person they are listening to, using an approach that incorporates and builds on aspects of the hierarchical frameworks and prototype-based mechanisms proposed within existing models of voice identity recognition. While the PPV model is more comprehensive than existing accounts of voice (identity) perception, its remit nonetheless has clear limits. For example, the current model has been proposed strictly within the framework of person perception from voices. However, we recognise that human person perception includes other sources and modalities of information, notably from faces and bodies. Other models of voice perception have explicitly depicted parallel/mirrored processing streams for face perception and allowed for interaction between these face and voice processing systems[9,11]. We, too, acknowledge this possibility and have, mainly for ease of visualisation and interpretation, restricted the present depiction and discussion of our model to voices only.

---

**Box 3 | The nature of voice-based representations**

Mental representations are assumed to be essentially categorical—the stored reference pattern for Person X is distinct from the reference pattern for Person Y (although they can be mapped onto a common acoustic voice space[77,79]). But how can categorical representations underpin the perception of dimensional characteristics like height, age, and perceived personality traits? Taking height as an example:

- Listeners may, on the one hand, have a representation of a person of average height, where the representation also encodes how height varies in the population. However, such an account cannot be readily incorporated into a mechanism where representations are stored as deviance patterns in relation to a prototype.
- Listeners might, on the other hand, have separate representations for a 'tall' voice and a 'small' voice. Listeners may consequently only recognise cues to height if a voice sounds particularly tall or small to them. Here, cues for people of average height are not salient and may thus not be recognised (but perhaps inferred as 'normal').

Other questions arise, for example, around what the nature of representations of characteristics that include possible sub-categories (e.g., American → New Yorker → Person from Queens) may be.

- Do we have entirely independent representations for all the available levels of specificity, or are representations partially co-dependent? That is, would hearing a voice that sounds American will automatically co-activate 'New Yorker' among other relevant sub-categories?
- Conversely, can listeners perceive 'Person from Queens' without having to first perceived 'American' and/or 'New Yorker'? The answer to this question will likely depend on how familiar a listener is with the different American accents (see also the direct vs indirect route to identity-specific perception in the PPV model).

These are just two examples of how the nature of mental representations is still relatively poorly understood for voice perception. It is a challenge for future research to uncover more about the nature and architecture of the representations that underpin person perception from voices and beyond.

---

Similarly, other types of information, related to speech and affective states are encoded in voices, and their perception has already been modelled in other accounts (e.g. refs. [9,11]). As acknowledged in those models, these parallel sources of information are likely to intersect with person perception from voices: A person might have a voice quality that does not immediately suggest a dominant personality, but the words they use might quickly compel a listener to think otherwise. These potential influences on person perception are currently not accounted for in the PPV model. This is not to suggest that we consider these intersections and interactions to be unimportant. On the contrary, establishing the extent to which these processes are mutually dependent—in function, time, space (e.g. neurobiological location), and mechanism—will be crucial[68–71].

With a view to future work, we hope that the PPV model presents the opportunity for greater inter-disciplinary synergy in testing its proposals, by taking a broader perspective on person perception from voices and thus capturing research questions from hitherto disparate empirical traditions. There is much work to be done to, for example, explore which and how different person characteristics are represented, which representations are processed and prioritised under which circumstances, how the perception of different person characteristics may interact and inform each other, and how precisely recognition is, in the end, mechanistically achieved for a range of different person characteristics. To answer these and similar research questions, studies will need to examine how the perception of multiple characteristics unfolds and is affected by experimental manipulations and situational contexts. We intend that the PPV model will provide a conceptual or theoretical backdrop to inform such studies.

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

## Acknowledgements

This work was supported by a Sir Henry Wellcome Fellowship awarded to Nadine Lavan (220448/Z/20/Z) and a Research Leadership Award from the Leverhulme Trust (RL-2016-013) awarded to Carolyn McGettigan.

## Author contributions

N.L. and C.M. have contributed equally to all aspects of this manuscript.

## Competing interests

The authors declare no competing interests.

**Additional information**

