## [Peer Review File · Communications Psychology]

24th Feb 23

Dear Nadine,

Thank you for your patience during the peer-review process. Your manuscript titled "A model for person perception from voices." has now been seen by 3 reviewers, and I include their comments at the end of this message.

The reviewers are overall enthusiastic about your work. However, they also mention a number of detailed issues that will need to be addressed. Before we can move towards publication of your manuscript in *Communications Psychology*, we would like to consider your response to these concerns in the form of a revised manuscript.

A Perspective article leaves room for the presentation of novel, perhaps controversial models. Nonetheless, we ask you to engage with the constructive criticism and requests for clarification offered by the referees. In addressing these concerns, please be transparent in the text about speculative aspects of your proposal, or those that are open to strong alternative viewpoints.

In sum, we invite you to revise your manuscript taking into account all reviewer comments.

EDITORIAL POLICIES AND FORMATTING

You will find a complete list of formatting requirements following this link:
<https://www.nature.com/documents/commsj-style-formatting-checklist-review-perspective.pdf>

Please use the checklist to prepare your manuscript for resubmission.

* **TRANSPARENT PEER REVIEW:** *Communications Psychology* uses a transparent peer review system. This means that we publish the editorial decision letters including Reviewers' comments to the authors and the author rebuttal letters online as a supplementary peer review file. We publish these records for all accepted manuscripts. However, on author request, confidential information and data can be removed from the published reviewer reports and rebuttal letters prior to publication.

If you have any questions about any of our policies or formatting, please don't hesitate to contact me.

Please use the following link to submit your revised manuscript and a point-by-point response to the referees' comments (which should be in a separate document to any cover letter):

[link redacted]

** This url links to your confidential home page and associated information about

manuscripts you may have submitted or be reviewing for us. If you wish to forward this email to co-authors, please delete the link to your homepage first **

We hope to receive your revised paper within 12 weeks; please let us know if you aren't able to submit it within this time so that we can discuss how best to proceed. If we don't hear from you, and the revision process takes significantly longer, we may close your file.

Please do not hesitate to contact me if you have any questions or would like to discuss these revisions further. We look forward to seeing the revised manuscript and thank you for the opportunity to review your work.

Best wishes,
Marike

Marike Schiffer, PhD
Chief Editor
Communications Psychology

REVIEWERS' EXPERTISE:

Reviewer #1: auditory/voice processing, recognition

Reviewer #2: auditory/voice processing, recognition

Reviewer #3: auditory/voice processing, recognition

REVIEWERS' COMMENTS:

Reviewer #1 (Remarks to the Author):

This is a well-written review which integrates existing literature on differences between familiar and unfamiliar voice processing, while proposing a novel potential 'common mechanism' for processing all voices i.e., processing 'who' is talking. This 'who' goes beyond the well documented identity recognition literature, expanding the scope to voice characteristics such as age and trustworthiness, all aspects which help form a rich holistic representation of the heard voice (potentially regardless of familiarity). The authors propose a new model for how these 'who' characteristics may be processed. This model (Figure 3) proposes that we may apply prototype processing to all voice characteristics, including, but not limited to identity. I commend the authors on this novel expansion. Below I list major and minor comments.

Major comments

My main concern is the 'common mechanism' for voice processing proposed by the authors, regardless of the voice familiarity. Specifically:

1) On page 3, the authors state: "we argue that instead of having distinct mechanisms for familiar vs unfamiliar identity perception, person perception from all voices is better characterised via a common mechanism involving the recognition of different person characteristics, be they identity-specific (for familiar voices) or not." Could the authors

elaborate on whether they consider that the perception of voice characteristics proceeds in parallel to identity processing? Depending on the familiarity, how might these other 'who' aspects be processed? The authors refer to this within the main text, but within the model it is not clear if, how, and when these differences emerge in relation to Figure 3A – prototype processing.

2) Page 16. "In existing models for the recognition of specific identities, there is only one possible match between a given voice and a stored representation of a specific person's identity. In contrast, the PPV model proposes the recognition of multiple person characteristics at the same time (or closely staggered)." But this does not seem to be reflected in Figure 3b – where it appears that the only thing that is processed is familiarity for a known voice i.e., the box with 'Familiar person(a)'. Do the authors mean that these characteristic elements do not continue to be extracted and compared for familiar voices? In this case how is a common mechanism proposed for both familiar and unfamiliar voice processing?

If the authors suspect that these representations of characteristics are already established for familiar voices (Figure 3b), this is not clear. Or do the authors expect a different weighting of the prototype processing (for all person characteristics) depending on the level of familiarity? (i.e., differences in Figure 3a). My uncertainty in this may be also relate to the use of terminology: "person-specific, persona-specific, and/or person-general characteristics" it would be good to explicitly define what is meant by these terms and keep it consistent in the text and when visualising/outlining the model.

3) Page 14. "We do not exclude the possibility that multiple prototypes may be available to better specify the deviant features of interest (see also below) but note that the existence of numerous, well-specified prototypes partially undermines the processing efficiencies highlighted in classic prototype models (e.g. 63)." Might the authors consider there might be differences in the prototypes for identity and for other characteristics. For example, with identity one wants to recognise the identity across variations in utterance (mean prototype, plus potential 'voice-space' around that identity), while for other voice characteristics one wants to recognise this despite variations in the identity. Do the authors have any comments on how these two processes may interact, in terms of their model (Figure 3).

4) Comments on Figures. General comment - It would be beneficial to have a detailed legend for each figure which describes the model and the visuals.

Figure 1. A legend which describes the stages and pathways of processing and what is meant by different arrow types e.g., filled v. dashed would be helpful.

Figure 2. A description of whether the voices were unfamiliar would be useful.

Figure 3a. Is this process identical for familiar and unfamiliar voices? Can authors please clarify in the legend and visual. "A model of Person Perception from Voices (PPV). a)

Illustration of how recognition of a person(a) or person characteristic may be mechanistically achieved, adapted from Lavner and colleagues" Does this mean recognition of a person (i.e., identity), persona, and person characteristics? It is a bit confusing the use of person(a) whether this relates to two concepts.

Please also detail what each of the arrows mean in the processing steps. The 'Voice Prototype' and 'Relevant Reference Pattern' appear at the same stage of processing. In the

original model, the deviating features (based on prototype comparison) were compared to reference patterns at the next step. How do these two steps take part simultaneously in this model?

In Figure 3a. There is only one 'Voice Prototype' – if the processing of multiple characteristics is happening potentially in parallel, might there be multiple prototypes accessed? This is discussed in the main text, but might be worth visualising the possibility.

5) Page 18. "Given these considerations, it may be that recognition of certain person characteristics can happen earlier than others that are less clearly marked (e.g., sex perception from adult voices being quicker than sex perception from child voices⁷²)". A temporal component in processing characteristics from the voice is suggested in the main text. However, in Figure 3b this is not understandable from the visuals. Can the authors add the potential for a temporal component to the visualisation e.g., timeline.

Minor comments

6) The authors define 'who' differently at times in the manuscript. It appears to represent all voice characteristics early in the paper and then later to apply specifically to identity e.g., page 6 "models thus only consider one specific case in which listeners are trying to make sense of who they are talking to (i.e., identity-specific perception)". It is a bit confusing, would be good to clarify exactly what is meant by 'who'. It would also be good to clarify early on in the manuscript what the definition of 'familiar' is.

7) On page 5 the authors note: "These studies showed a double dissociation, where individuals who had difficulties in voice identity processing (known as phonagnosia) following a brain injury were able to successfully complete identity discrimination tasks with unfamiliar voices but could not recognise familiar voices, and vice versa^{4,13,14} (see also Garrido and colleagues' report of developmental phonagnosia¹⁵)." Similar observations have also been reported by Roswadowitz et al., 2014, 2017 in developmental phonagnosia and autism by Schelinski et al., 2017.

8) Line 595. Also Maguinness et al., 2018 have a mirrored face part to their voice model.

Reviewer #2 (Remarks to the Author):

This is a relevant, timely, and compelling conceptual framework, which will significantly contribute to the field of voice perception. The proposed model aims to offer a broader perspective on person perception from voices beyond identity recognition. I would therefore like to congratulate the authors for their thorough and critical review of existing literature and conceptual models on voice identity perception, and, more importantly, for their comprehensive and well-thought-out model that goes beyond the state of the art. The manuscript is exceptionally well written; the main arguments are clearly formulated and presented in a convincing manner. This manuscript will certainly be of interest to the readers of Communications Psychology and, in my opinion, should be accepted for publication. I have no suggestions for improvement but only a minor comment:

-On p. 18, references should be added to support the following statements: "For example, pitch cues to an adult's sex can be perceived within a few glottal cycles, while information about a vocal tremor indexing old age or poor health requires longer exposure."; "...voice pitch in humans is a highly salient and reliable cue to differences between adult male and female voices, while it is less reliable for children".

Reviewer #3 (Remarks to the Author):

This is an interesting and manuscript that describes a novel theoretical model for the perception of speaker attributes in both familiar and unfamiliar voice identities. The literature is well reviewed and presented, and the manuscript gives much food for thoughts (I would personally recommend it to my students and collaborators). The manuscript is well written.

I have only two major remarks about the details of the model.

1. The person perception model relies heavily on prototype matching/comparison mechanisms. It would be relevant to have more considerations about how the model can account for the perception of continuous person attributes such as age or emotional intensity (arousal). Is the perception of continuous attributes based on the comparison to prototypes stored for the extremes of the continua? If the model doesn't account for the perception of continuous attributes the authors should instead describe this as a limitation. Alternatively, it would be very useful if the authors could sketch an alternative model that does not rely exclusively on prototype matching mechanisms.

2. The authors propose that heard voice representations are compared with one single voice prototype for the computation of the deviant features. Why the authors opt for one single prototype is unclear, and should be better justified. At a second stage, deviant features are compared to reference patterns. It is not clear how these reference patterns differ functionally from a prototype, and why this particular architecture involving two comparison stages is more plausible than an architecture that directly compares the relevant acoustic features with a set of (possibly hierarchically organized) prototypes. I understand numerous prototypes might "the processing efficiencies [...] in prototype models". However, in the absence of more detailed proposals about the mechanisms involved, it is plausible to hypothesize that a system with e.g., hundreds of prototypes still works efficiently.

Detailed remarks:

L[ine] number: [original text] comment/proposed edit

L:184 [On the other hand,,] delete extra comma

L218: [It is again not the case that unfamiliar voices cannot be emotionally evocative] It would perhaps facilitate the reader to have the double negation explicitly resolved into a positive statement.

L221: [These situations are, however, likely to be exceptions, and not the rule, for unfamiliar voice perception.]: I am not sure I agree. It isn't uncommon, for example, to judge a never heard before voice as unpleasant, or to get worried if we hear a nearby scream. Perhaps this sentence can be safely removed?

Fig. 3a: it would help if the panel made explicit that the processing chain applies to both familiar and unfamiliar voices.

Fig. 3b: does the first box to the right of the voice signal essentially represent Fig. 3a? If yes the link should be clarified and the voice signal in this panel should perhaps not be represented.

L395: [each box with a solid outline describes a recognised percept, and each box with a dashed outline describes a computational process] isn't the leftmost box of 3b a computational process? How are the voice prototype and the relevant reference pattern in 3a a percept?

L417: [We cautiously] why cautiously?

L478: [For this direct route, following the extraction of person-relevant features from an incoming voice signal, the common recognition mechanism matches the analysed voice signal (i.e., pattern of deviant features) directly with a stored representation of a known person.] Isn't this inconsistent with your proposal above of a single average voice prototype?

L547: [The multivariate nature] perhaps multi-attribute would be more appropriate?

L602: [.. As acknowledged] remove extra dot

REVIEWERS' COMMENTS:

Comment 0 (relevant to comments from Reviewer 1 and 3): In some of the reviewer comments below, there is some conflation of the concepts of “prototypes” vs “representations. To clarify how prototype and representations differ from one another, functionally and in terms of their nature, we will first review statements from the paper that originally proposed a model of prototype-based coding for voice (identity) recognition (Lavner et al., 2001) in this rebuttal letter.

For prototypes, Lavner et al. (2001) state: *“Our prototype model assumes the presence of a prototype (‘average’) voice (or several prototypes) in each listener’s memory. This prototype pattern is comprised of an ensemble of acoustic features, related to the language, the accent, the phonemes and allophones, and to the voice production system. The prototype voice could reflect the average of speakers’ features or a very common voice.”*

For representations (or ‘stored reference patterns’ in Lavner et al.’s (2001) terminology), the paper states: *“For each new voice, only those features that significantly deviate from the prototype are stored (memorized) for the long term, and identification of familiar voices is based on searching and locating the voice, using only those features deviating from the prototype.”*

A prototype is thus an average(d) voices, while a representation (of a specific voice identity, for example) is the pattern of deviations from the prototype. This deviance-based mechanism is not unique to this paper. Similar functional distinctions between prototypes and representations can also be found in, for example, in the face perception literature (Bruce & Valentine, 1986; Light et al., 1979).

Prototype models for voice and face (identity) recognition, in general, have – as far as we can reconstruct this from the existing literature – been originally proposed to account for the empirical finding that distinctive voices and faces, i.e., voices/faces that deviate most from an average or prototypical voice/faces, are better recognised than less distinctive voices and faces in terms of their specific identities.

We now briefly explain and evaluate this history of prototype-based models in Box 3 in the manuscript, which was added to address some of the reviewer comments in the letter below. Specifically, Lavner et al., (2001) state: *“The main findings of the psychoacoustic experiments support the hypothesis that speaker identification depends on the deviation of the acoustic features of the speaker’s voice from an estimated average template. The stronger the deviation of the acoustic features of a certain speaker from the population average, the easier it would be to identify him or her; the less the features deviate from the population template, the lower the chances that a speaker will be correctly identified.”*

In light of the confusion about the specifics of the prototype mode, we have for the revised manuscript more clearly defined these terms. We now also provide an alternative mechanism for recognition processes that is not functionally-reliant on prototypes. Please see the responses below for additional details.

Reviewer #1 (Remarks to the Author):

Summary Comment: This is a well-written review which integrates existing literature on differences between familiar and unfamiliar voice processing, while proposing a novel potential ‘common mechanism’ for processing all voices i.e., processing ‘who’ is talking. This ‘who’ goes beyond the well documented identity recognition literature, expanding the scope to voice characteristics such as age and trustworthiness, all aspects which help form a rich holistic representation of the heard voice (potentially regardless of familiarity). The authors propose a new model for how these ‘who’ characteristics may be processed. This model (Figure 3) proposes that we may apply prototype processing to all voice characteristics, including, but not limited to identity. I commend the authors on this novel expansion. Below I list major and minor comments.

Summary Response: Thank you for the overall positive assessment of our paper!

Major comments

Comment 1: My main concern is the ‘common mechanism’ for voice processing proposed by the authors, regardless of the voice familiarity. Specifically:

1) On page 3, the authors state: “we argue that instead of having distinct mechanisms for familiar vs unfamiliar identity perception, person perception from all voices is better characterised via a common mechanism involving the recognition of different person characteristics, be they identity-specific (for familiar voices) or not.” Could the authors elaborate on whether they consider that the perception of voice characteristics proceeds in parallel to identity processing? Depending on the familiarity, how might these other ‘who’ aspects be processed? The authors refer to this within the main text, but within the model it is not clear if, how, and when these differences emerge in relation to Figure 3A – prototype processing.

Response 1: As the reviewer notes, we discuss in the text that the processing of voice characteristics vs person(a) characteristics might unfold serially or in parallel, dependent on voice familiarity (see also Response 3). However, given the lack of empirical evidence testing these kinds of proposals, we have on purpose not comprehensively specified in our model – or in our figure – how exactly serial or parallel processing may work (see also Comment 4 below). Our overview figure of the PPV model (formerly Figure 3(b), now Figure 3) illustrates via arrows some of the *possible* interactions, for example, between recognition of a familiar person(a) and other characteristics.

We furthermore now outline in the manuscript that independent as well as parallel processing of multiple recognition mechanisms are possible within our model:

P13: “In our model, following the extraction of person-relevant features from the acoustic voice signal, listeners recognise any number of person characteristics based on these features. In line with previous models, the PPV model indicates a possible route to account for recognition of familiar persons or personae (see below) and adds to this the capacity for recognition of one or more individual person characteristics (sex, age, health, etc.). Although visually depicted as two routes, these processes are not mutually exclusive to one another: While person-specific recognition may be prioritised depending on the listening situation, individual person characteristics may be recognised in parallel with recognising a familiar person/persona (see below for further considerations).”

P16: “Despite the possible interactions, the PPV model still allows for direct person/persona recognition via matching to a single stored identity representation as outlined above (adapted from Belin and colleagues⁹, Maguinness and colleagues³), where prior or parallel processing of other person characteristics is not obligatory.”

See also Box 2 for further consideration about serial and parallel processing.

We hope this addresses the reviewer’s comment.

Comment 2a: 2) Page 16. “In existing models for the recognition of specific identities, there is only one possible match between a given voice and a stored representation of a specific person’s identity. In contrast, the PPV model proposes the recognition of multiple person characteristics at the same time (or closely staggered).” But this does not seem to be reflected in Figure 3b – where it appears that the only thing that is processed is familiarity for a known voice i.e., the box with ‘Familiar person(a)’. Do the authors mean that these characteristic elements do not continue to be extracted and compared for familiar voices?

Response 2a: Before addressing the questions raised by the reviewer in relation to this quote from our paper, we note that the second sentence in the quote the reviewer highlights was primarily intended to describe how complex impressions from unfamiliar voices are formed (see the rest of the paragraph in the manuscript, copied below). In light of the comments made by the reviewer below, this seems to perhaps have not been clear, such that we have now clarified this in the revision:

P14: “In existing models for the recognition of specific identities, there is only one possible match between a given voice and a stored representation of a specific

person's identity, which represented by the direct route to the perception of a known person or persona in the PPV model. The PPV model proposes additionally the recognition of multiple person characteristics at the same time or closely staggered in time. This may happen on a regular basis when we encounter unfamiliar voices and derive complex, multi-attribute impressions from these voices. Emerging evidence from voice research suggests that this perception of different person characteristics from unfamiliar voices may indeed be structured."

As depicted by the arrows in Figure 3 and explained in the main text, the possibility that a highly familiar person might be recognised rapidly in one situation does not preclude the possibility that in another situation (unclear signal, unexpected context) recognition of that same person may not be achieved before other component characteristics have been recognised in parallel. Similarly, we do not rule out that listeners can process both person characteristics and e.g., the identity of a specific person in parallel (as is required by the listening situation).

Within the manuscript, we have clarified further how this section of Figure 3 should be interpreted:

P14: "The PPV model builds on previous model (See Section 2) and is therefore at its core a hierarchical framework (see^{10,12}), within which a common recognition mechanism operates (see Section 5.2) to account for the common goal of person perception from voices. In our model, following the extraction of person-relevant features from the acoustic voice signal, listeners recognise any number of person characteristics based on these features. In line with previous models, the PPV model indicates a possible route to account for recognition of familiar persons or personae (see below) and adds to this the capacity for recognition of one or more individual person characteristics (sex, age, health, etc.). Although visually depicted as two routes, these processes are not mutually exclusive to one another: While person-specific recognition may be prioritised depending on the listening situation, individual person characteristics may be recognised in parallel with recognising a familiar person/persona (see below for further considerations)."

Comment 2b: In this case how is a common mechanism proposed for both familiar and unfamiliar voice processing?

Response 2b: In Figure 3, each box (or Recognition Unit when borrowing the terminology from Belin et al., 2011 and Bruce & Young, 1986) may correspond to a percept of an individual familiar person, a familiar persona, or a person characteristic. We propose that the common mechanism to achieve these percepts is a *recognition* mechanism, upon which we elaborate in Figure 4 (previously Figure 3a) and in the accompanying text.

We, therefore, provide for the first time 1) a mechanistic account for how person characteristics (other than identity) may be perceived and 2) refocused the literature from proposing different mechanisms (discrimination vs recognition) to account for familiar vs unfamiliar identity perception to consider person perception more broadly, via an approach that considers how person-related information from voices may be processed outside of experimental tasks. This is a significant conceptual advance that has to our knowledge not been considered in the literature before.

Comment 2c: If the authors suspect that these representations of characteristics are already established for familiar voices (Figure 3b), this is not clear. Or do the authors expect a different weighting of the prototype processing (for all person characteristics) depending on the level of familiarity? (i.e., differences in Figure 3a).

Response 2c: Please see Comment 2a. Once a specific familiar person or perhaps persona has been recognised, listeners have access to all their existing knowledge about this person(a), which overlaps with information about other person characteristics such as (the typical) sex, age, etc. of the person(a). In Figure 3, the model therefore includes mutual transfer of information between the recognition process

and the “Cognitive System” that stores such memories and knowledge about known individuals (“That’s X. I know they’re a woman, 35 years old, tall”).

For a familiar voice, bottom-up recognition of these characteristics via Recognition Units may not take place if specific person recognition happens sufficiently rapidly/directly. However, this does not necessarily mean that these characteristics would not continue to be processed/monitored to some degree in a bottom-up manner. Similarly, if voices are less familiar, then the bottom-up recognition of some other person characteristics may routinely precede identity-specific recognition (see the second paragraph on P16 of the manuscript: “It is possible that an “indirect route” to (familiar) person or persona recognition via the recognition of other person characteristics may in fact be the typical means of recognising a familiar person or persona...”)

Thus, in short: In terms of the relative prioritisation of *representations* (whether coded in relation to a prototype or not) in the recognition process – yes, we would expect that top-down cognitive processes may play a role depending on factors such as the level of familiarity and listening situation, as specified on p 15/16 and Box 2 in the manuscript.

Comment 2d: My uncertainty in this may be also relate to the use of terminology: “person-specific, persona-specific, and/or person-general characteristics” it would be good to explicitly define what is meant by these terms and keep it consistent in the text and when visualising/outlining the model.

Response 2d: Person-specific representations correspond to the representations of specific familiar individuals. Persona-specific representations correspond to representations of specific familiar *types of people*, e.g. Valley Girl, City Banker. Person-general representations correspond to perceived characteristics of people, familiar or unfamiliar. In revision, we have thoroughly checked the manuscript to ensure that these terms have been clearly defined and used consistently.

P12: “Here, we reframe person perception as a process that has a common perceptual goal and which is achieved via a common mechanism of recognition of person-specific, persona-specific, and/or person-general (e.g., sex, age, etc.) characteristics.”

Comment 3: 3) Page 14. “We do not exclude the possibility that multiple prototypes may be available to better specify the deviant features of interest (see also below) but note that the existence of numerous, well-specified prototypes partially undermines the processing efficiencies highlighted in classic prototype models (e.g. 63).” Might the authors consider there might be differences in the prototypes for identity and for other characteristics. For example, with identity one wants to recognise the identity across variations in utterance (mean prototype, plus potential ‘voice-space’ around that identity), while for other voice characteristics one wants to recognise this despite variations in the identity. Do the authors have any comments on how these two processes may interact, in terms of their model (Figure 3).

Response 3: First, our paper makes a distinction between prototypes and representations, and it is not clear to which the reviewer to refer to here (see Comment 0). To clarify: A *prototype* was invoked in our manuscript as part of a specific recognition mechanism within which all incoming voice signals are compared with a prototype voice in the calculation of deviant feature patterns. These deviance patterns are then compared with stored deviance patterns - *representations* - of different person(a) identities and other person characteristics. This mechanism is directly adapted from the original work of Lavner et al. 2001 (and also aligns with the updated version of the model by Maguinness et al., 2018).

Here, the reviewer specifically asks whether different *prototypes* (e.g., identification versus age prototypes) might be different in nature. We did not propose that there are multiple prototypes within our framework, such that it is difficult to speculate further (see however, Reviewer 3, Comment 2 for some considerations around having multiple prototypes). When considering stored *representations*, of which we propose there are multiple, we do not suggest any qualitative difference in the nature of representations for identities versus other characteristics. Just as a listener needs to recognise a given identity across different utterances or speaking styles, so too do they need to recognise that specific person as distinct from other persons (i.e., processing both within- and between-person variability). This is the same for person characteristics: perceiving sex must be ideally robust to within-person variations (a man laughing – thus often dramatically increasing the F0 of their voice) as well as between-person

variability. Thus, all representations might be subject to the same functional pressures during person perception from voices.

One obvious difference in the nature of prototypes/representations for identity versus other characteristics may be associated with granularity: There are many different possible identities that a person might recognise, while there are only a relatively limited number of categories for 'gender', for example. At this point, we do not have the scope to specify what the implications of this are for the interaction of these different kinds of representations/percepts, although in the manuscript we offer some suggestions based on the published literature (e.g., sex perception precedes identity recognition; e.g. Dobs et al., 2019; Owren & Bachorowski, 2007). We hope this answers the reviewer's question.

Comment 4: 4) Comments on Figures. General comment - It would be beneficial to have a detailed legend for each figure which describes the model and the visuals.

Response 4: Thank you for this suggestion. We had added some clearer signposting within the figure illustrations and legends, although we refrain from overly detailed figure captions to avoid redundancy with the main text. For details, see the comments and responses below.

At this point, we thought it would be helpful to more clearly explain our general approach to the visualisations of our model, given several of the reviewer's comments concern the presentation of the figures. When extending existing models of voice identity recognition to encompass person perception more broadly, we needed to for the first time account for how recognition/representations of different kinds of person characteristics may be achieved and may interact. In comparison to models focused on identity recognition, it is there difficult to depict all possible components and their interactions of our more complex model within one figure. For some aspects, we therefore sought to avoid over-specifying the model in the absence of sufficient supportive empirical evidence, but also for ease of reading we wished to minimise visual complexity of the different figures, while still conveying core concepts. That said, we agree with the reviewer that there were some ambiguities in the figures (and text). We therefore have in revising the paper made attempts to set out the text and visualisations in a more streamlined fashion that we hope will reduce some ambiguities that may have caused confusion. In general, we are also happy to revise our figures further if we have missed anything the reviewer considers to be important in the revision.

Comment 5: Figure 1. A legend which describes the stages and pathways of processing and what is meant by different arrow types e.g., filled v. dashed would be helpful.

Response 5: Thank you for this suggestion. The different stages of the model are (very briefly) outlined and described in the main text. The main point of this review of existing models is to delineate the scope and remit of the models, rather than give a detailed account of how each model works. Thus, we have for now refrained from adding a longer description of the pathways/stages to the figure legend. If this is deemed necessary, we are, however, happy to add this.

The different arrow types in Figure 1 were directly taken from the figures included in the original papers. However, we appreciate in retrospect that in our adapted Figure 1, there was no need for the different line types. We have now aligned line types for all arrows, such that no further explanation of their meaning is required.

Comment 6: Figure 2. A description of whether the voices were unfamiliar would be useful.

Response 6: Thank you. The voices were indeed unfamiliar and we have added this detail to the figure legend of Figure 2.

Comment 7a: Figure 3a. Is this process identical for familiar and unfamiliar voices? Can authors please clarify in the legend and visual. "A model of Person Perception from Voices (PPV). a) Illustration of how recognition of a person(a) or person characteristic may be mechanistically achieved, adapted from Lavner and colleagues" Does this mean recognition of a person (i.e., identity), persona, and person characteristics? It is a bit confusing the use of person(a) whether this relates to two concepts.

Response 7a: The reviewer is correct and we have adjusted the wording to make this more explicit (now in Figure 4). Although person and persona recognition are conceptually very similar, they are mechanistically likely no different to person characteristic recognition in our model and we have therefore now spelled out each concept separately throughout the manuscript.

Comment 7b: Please also detail what each of the arrows mean in the processing steps. The 'Voice Prototype' and 'Relevant Reference Pattern' appear at the same stage of processing. In the original model, the deviating features (based on prototype comparison) were compared to reference patterns at the next step. How do these two steps take part simultaneously in this model?

Response 7b: Apologies. This was an error in redrawing the Lavner et al. (2001) figure. The figure has been adjusted accordingly for this revision and now appears as Figure 4(a). A description of the model can be found in the main text (P18).

Comment 8: In Figure 3a. There is only one 'Voice Prototype' – if the processing of multiple characteristics is happening potentially in parallel, might there be multiple prototypes accessed? This is discussed in the main text, but might be worth visualising the possibility.

Response 8: In response to the reviewer's specific request regarding a visual illustration of multiple prototypes: Given that we do not propose a multi-prototype mechanism (although we acknowledge that, in principle, there could be multiple prototypes in the text, see P18 and Box 3), we refrain from depicting the possibility of multiple prototypes on the figures. See also Response 3 above.

Comment 9: Page 18. "Given these considerations, it may be that recognition of certain person characteristics can happen earlier than others that are less clearly marked (e.g., sex perception from adult voices being quicker than sex perception from child voices⁷²)". A temporal component in processing characteristics from the voice is suggested in the main text. However, in Figure 3b this is not understandable from the visuals. Can the authors add the potential for a temporal component to the visualisation e.g., timeline.

Response 9: We had considered adding a timeline to the figure but decided against it, since this would perhaps suggest linear passing of time, for which we have no conclusive empirical evidence or predictions based on the extant literature. We also note that other hierarchical models (see Belin et al., 2004) also imply a temporal procession of stages but do not add a timeline to the models.

Minor comments

Comment 10: 6) The authors define 'who' differently at times in the manuscript. It appears to represent all voice characteristics early in the paper and then later to apply specifically to identity e.g., page 6 "models thus only consider one specific case in which listeners are trying to make sense of who they are talking to (i.e., identity-specific perception)". It is a bit confusing, would be good to clarify exactly what is meant by 'who'. It would also be good to clarify early on in the manuscript what the definition of 'familiar' is.

Response 10: Thank you. We have gone through the paper and checked that our references to "who" refers to a broader definition of "who a person may be, based on any characteristics". The sentence highlighted by the reviewer on page 6 indeed used "who" in a more specific sense, which we have now clarified.

P5: "The models thus only consider one case in which listeners are trying to make sense of which specific person they are talking to (i.e., identity-specific perception)"

Comment 11: 7) On page 5 the authors note: "These studies showed a double dissociation, where individuals who had difficulties in voice identity processing (known as phonagnosia) following a brain injury were able to successfully complete identity discrimination tasks with unfamiliar voices but could not recognise familiar voices, and vice versa^{4,13,14} (see also Garrido and colleagues' report of developmental phonagnosia¹⁵)." Similar observations have also been reported by Roswadowitz et al., 2014, 2017 in developmental phonagnosia and autism by Schelinski et al., 2017.

Response 11: Thank you - we have now added these references relevant to phonagnosia.

Comment 12: 8) Line 595. Also Maguinness et al., 2017 have a mirrored face part to their voice model.

Response 12: Apologies for this oversight. We have now also added this reference to the manuscript.

Reviewer #2 (Remarks to the Author):

Summary Comment: This is a relevant, timely, and compelling conceptual framework, which will significantly contribute to the field of voice perception. The proposed model aims to offer a broader perspective on person perception from voices beyond identity recognition. I would therefore like to congratulate the authors for their thorough and critical review of existing literature and conceptual models on voice identity perception, and, more importantly, for their comprehensive and well-thought-out model that goes beyond the state of the art. The manuscript is exceptionally well written; the main arguments are clearly formulated and presented in a convincing manner. This manuscript will certainly be of interest to the readers of Communications Psychology and, in my opinion, should be accepted for publication. I have no suggestions for improvement but only a minor comment:

Summary Response: Thank you for the positive assessment of our model!

Comment 1: -On p. 18, references should be added to support the following statements: "For example, pitch cues to an adult's sex can be perceived within a few glottal cycles, while information about a vocal tremor indexing old age or poor health requires longer exposure."; "...voice pitch in humans is a highly salient and reliable cue to differences between adult male and female voices, while it is less reliable for children".

Response 1: Thank you. We have added supporting references to the manuscript.

Reviewer #3 (Remarks to the Author):

Summary Comment: This is an interesting and manuscript that describes a novel theoretical model for the perception of speaker attributes in both familiar and unfamiliar voice identities. The literature is well reviewed and presented, and the manuscript gives much food for thoughts (I would personally recommend it to my students and collaborators). The manuscript is well written.

Summary Comment: Thank you for this positive summary of our paper.

I have only two major remarks about the details of the model.

Comment 1a: 1. The person perception model relies heavily on prototype matching/comparison mechanisms. It would be relevant to have more considerations about how the model can account for the perception of continuous person attributes such as age or emotional intensity (arousal). Is the perception of continuous attributes based on the comparison to prototypes stored for the extremes of the continua? If the model doesn't account for the perception of continuous attributes the authors should instead describe this as a limitation.

Response 1a: Thank you for this comment.

It is not entirely clear whether the reviewer makes the same distinction as we do between prototypes and representations. To clarify: A *prototype* was invoked in our manuscript as part of a specific recognition mechanism within which all incoming voice signals are compared with a prototype voice in the calculation of deviant feature patterns (see Comment 0). This mechanism is taken directly from Lavner et al. 2001 (see also Maguinness et al., 2018), which is the original paper proposing a prototype-based model of voice perception. These deviance patterns are then compared with stored reference patterns - *representations* - of different person identities, personae, and person characteristics.

In our original manuscript, we invoked only one prototype, where deviance patterns emergent from comparison with this prototype could be used in matching with multiple stored representations. While this prototype-based approach is taken from the existing literature, we acknowledge that this mechanism and its distinction between the (single) prototype and (multiple) representations appears needlessly complicated (see also Response 2 below). An alternative view, which we now consider more carefully in the revised paper, is that there may be no need for representations to take the form of deviance patterns at all, which in turn may remove the need for prototypes entirely (see Figure 4(b) and accompanying text; see Response 1b below).

With our definitions in mind, and in response to the reviewer's question: Consideration of person characteristics that can be judged in a continuous fashion might be represented and recognised within our model is an issue that we also grappled with in writing the paper. In Box 4, we have discussed how the PPV model may account for continuous characteristics (see below). We believe that this text covers the issue highlighted by the reviewer.

From the existing text of Box 4:

P15: "For example, we could assume that categorical representations of continuous characteristics would need to encapsulate the dimensional nature of the characteristics in question. Listeners may, for example, have a representation of a person of average height, where the representation also encodes how height varies in the population. However, such an account cannot readily be incorporated into the mechanism outlined by prototype-based models of person recognition. Alternatively, listeners might need to access separate representations for a "tall" voice and a "small" voice. In a height estimation task, the listener's response would then reflect a match with "tall", a match with "small", or no match (i.e. response around the known average height). Outside of experimental tasks, listeners may indeed only recognise cues to height if a voice sounds particularly tall or small to them. Cues for people of average height are not salient and may thus not be recognised (but perhaps inferred as 'normal')."

We would be happy to expand further on this, but believe that there is currently no empirical data or theoretical work available to further flesh out or substantiate these broad hypotheses.

Comment 1b: Alternatively, it would be very useful if the authors could sketch an alternative model that does not rely exclusively on prototype matching mechanisms.

Response 1b: We revisited this issue in more detail when revising the paper and have made considerable changes to the text to update our thinking. Initially, we proposed a prototype-based recognition process primarily on the basis that it is as far as we are aware the only existing mechanistic model of voice identity recognition (as outlined by Lavner et al. (2001) and Maguinness et al. (2018)). However, when more carefully thinking about the plausibility and when seeking the empirical basis for including prototypes in these earlier models, we could not find a comprehensive account in either paper. From our wider reading, including the face perception literature (e.g., Bruce & Valentine 1986, Valentine 1991), the clearest justification we can see for invoking prototypes and calculating deviance patterns is to account for empirical observations that face and voice identities that are highly deviant from the prototype/average are perceived as distinctive and are more accurately recognised. Thus, representing person identities in terms of their deviance from the prototype/norm is a plausible means by which distinguishing features of an individual might be made more perceptually salient – perhaps a kind of signal-to-noise amplification (see also Comment 0). However, there may be alternative explanations for why deviant identities are distinctive – for example, voices with properties far from the prototype/"norm" may also lie in less populated parts of the notional voice space where there are fewer competitors (but where listeners also have less experience, thus forming unstable long-term memories of these voices; Papcun et al., 1988).

Thus, in our revised paper we now outline an additional, alternative mechanism that removes the need for prototypes and deviance patterns, and instead accounts for recognition – of persons, personae, and person characteristics – by directly matching whole incoming feature patterns with (similarly whole) stored pattern representations (as mentioned above by the reviewer). Specifically, we now include as Figure 4(b) this alternative, prototype-free model and discuss prototypes and their possible role in more

detail within a new Box 3 and on P20/21 in the main text. Resolving between these two alternatives is not possible at the moment but should be the focus of future empirical work.

Comment 2: 2. The authors propose that heard voice representations are compared with one single voice prototype for the computation of the deviant features. Why the authors opt for one single prototype is unclear, and should be better justified. At a second stage, deviant features are compared to reference patterns. It is not clear how these reference patterns differ functionally from a prototype, and why this particular architecture involving two comparison stages is more plausible than an architecture that directly compares the relevant acoustic features with a set of (possibly hierarchically organized) prototypes. I understand numerous prototypes might “the processing efficiencies [...] in prototype models”. However, in the absence of more detailed proposals about the mechanisms involved, it is plausible to hypothesize that a system with e.g., hundreds of prototypes still works efficiently.

Response 2: We agree with the reviewer that there were key details that were underspecified in our paper with regard to prototype-based coding. This is notably also the case for the literature more generally (see our comment in the original manuscript that “There are therefore considerations around the nature, number and use of prototypes that are currently underspecified in prototype models of voice (and face) perception.”). In Response 1b above, we outline how our revised paper more directly tackles the challenges of whether prototypes are functionally necessary to explain person perception, now including an additional, alternative mechanism based on matching to stored representation patterns in the absence of prototype-based deviance coding.

In the current response, we turn to the issue of one vs many prototypes, and how this relates to differences between prototypes and reference patterns (which we also refer to as *representations*). Again, we have wrangled with these issues but hope that our account below and in the revised paper will explain our thinking with greater clarity.

In our view of the relevant literature - which we again note is focused on accounting for familiar voice/face identity recognition only - prototype-based (or norm-based) coding for identity recognition is invoked in order to generate deviance patterns that amplify the “signal” against the noise and explain behavioural findings of superior recognition for distinctive identities. In a prototype-based account of voice identity recognition, the prototype represents something like an average of all voices, such that the process of subtracting it away strips away the most redundant and uninformative cues for recognition. But generating a deviance pattern is not in itself sufficient for recognition – the pattern needs to be matched to some sort of stored representation. In prototype-based models of identity processing, such representations of known identities correspond to the stored reference patterns. According to the previous literature, this is the baseline distinction between a prototype and a reference pattern / representation: the former is used to calculate deviance patterns, and the latter is used for matching-based recognition. Crucially, it is only within such a two-stage model that we require these 2 different terms.

Keeping in mind this definition of prototype, the notion of multiple prototypes for person perception (e.g. “tall” prototype, “elderly” prototype) to our mind requires us to consider the mechanistic plausibility of deviance patterns for the purpose of recognising other kinds of person characteristics. We have struggled to see how this could work. First, applying the logic of the prototype-based identity recognition models (e.g. Lavner et al., 2001), it is not intuitive to conceptualise how a voice that is very deviant from a prototypical “female” would be better recognised as an example of a woman (as opposed to a child or an adult of another gender). That is, distance from a prototype *per se* does not intuitively account for better recognition on any recognisable characteristic beyond perhaps a unique identity. Second, adding an option for selecting one of multiple prototypes to an already complex mechanism, without there being a clear functional advantage seems counterproductive (see also Box 3, which we added to the currently manuscript to – more briefly – highlight some of these issues).

If we instead step back from the two-stage model as the reviewer suggests, prototypes and representations become redundant both conceptually and mechanistically. Here, any incoming signal is compared directly with a representation – we outline this possibility visually in Figure 4(b) of the revised manuscript. In a framework where prototypes and representations are equivalent, we would

agree with the reviewer that there may exist of a large number of stored feature patterns coding for different persons/personae/characteristics, where an incoming signal could be matched to any number of these, and where there would be possibilities for hierarchical and interactive recognition of multiple representations both in serial and parallel. However, for explanatory consistency, we would not refer to these representations as *prototypes*, so as not to invoke any kind of prototype-based coding. We note, of course, that this particular account fails to explain some of the empirical findings suggesting a perhaps functional role for prototype-/norm-based coding (e.g. Lavner et al., 2001).

We hope this explanation, and the revisions to the paper (see above), are helpful. Although invoking a (single-)prototype-based coding principle for recognition seemed superficially to be the most plausible option in our initial account, both in relation to existing models and our overall purpose, a deeper consideration of these mechanistic components has given us greater clarity on the limitations of two-stage accounts in general, and multi-prototype versions of these in particular.

Additional Comments:

Detailed remarks:

L[ine] number: [original text] comment/proposed edit

L:184 [On the other hand,,] delete extra comma

Thank you. This has been corrected.

L218: [It is again not the case that unfamiliar voices cannot be emotionally evocative] It would perhaps facilitate the reader to have the double negation explicitly resolved into a positive statement.

This has been corrected:

P7: “However, unfamiliar voices can still be emotionally evocative as a result of salient personal memories as they may be present for familiar voices: For example, we may experience an emotional reaction when hearing someone speak in an accent that we have not heard since childhood, or hearing someone who *sounds like* a familiar individual.”

L221: [These situations are, however, likely to be exceptions, and not the rule, for unfamiliar voice perception.]: I am not sure I agree. It isn't uncommon, for example, to judge a never heard before voice as unpleasant, or to get worried if we hear a nearby scream. Perhaps this sentence can be safely removed?

Thank you for this comment. We agree with the reviewer that unfamiliar voices can be perceived as unpleasant, etc. However, with this sentence, we were trying to examine personal emotional salience as it might be found for familiar voices. We would argue these kinds of experiences in response to unfamiliar voices are possible, but truly an exception to the rule.

We hope the revised text clarifies our point:

P7: “However, unfamiliar voices can still be emotionally evocative as a result of salient personal memories as they may be present for familiar voices: For example, we may experience an emotional reaction when hearing someone speak in an accent that we have not heard since childhood, or hearing someone who *sounds like* a familiar individual.”

Fig. 3a: it would help if the panel made explicit that the processing chain applies to both familiar and unfamiliar voices.

This has been updated in the paper within what is now Figure 3.

Fig. 3b: does the first box to the right of the voice signal essentially represent Fig. 3a? If yes the link should be clarified and the voice signal in this panel should perhaps not be represented.

In the original paper, we adapted the visualisation of the voice/speech signal from Lavner et al. (2001) and Maguinness et al. (2018), which outlines a voice (identity) recognition mechanism from a structural analysis of an incoming waveform / voice signal to the process of identification. Similarly, our figure of the PPV model charted a pipeline for person perception from voices that proceeded from the incoming acoustic signal, through structural analysis and recognition, to the engagement of the wider cognitive system – this echoes the hierarchical structure of face and voice models such those described by Bruce & Young (1986) as Belin et al. (2004).

As the reviewer perhaps anticipates in their comment, the specific function of this particular box is not well-described: Lavner et al. (2001) do not seem to describe the nature of this box at all, Maguinness et al. (2018) describe it as “the perceptual level of processing, where identity-features are extracted and merged to create a coherent voice percept i.e. identity-feature analysis”. We tend to agree with Maguinness et al (2018) and further believe that this box closely maps onto to the “voice structural analysis” box in Belin et al.’s (2004) model. This has now been clarified by adding labels from the Belin et al. (2004) model to our model (see Figure 3).

L395: [each box with a solid outline describes a recognised percept, and each box with a dashed outline describes a computational process] isn't the leftmost box of 3b a computational process? How are the voice prototype and the relevant reference pattern in 3a a percept?

Thank you for pointing this out. This observation is correct and it has helped us to identify a number of inconsistencies in the labelling of boxes within the original submitted manuscript. We have revised the figures and their accompanying legends to remove these inconsistencies, hopefully making the component parts of the model clearer.

P18: “Across panels a) and b), each box with a solid outline describes a recognised percept, each box with a dashed outline describes a computational process, and each box with a dotted line describes a type of stored cognitive object (e.g., a representation) supporting recognition.”

L417: [We cautiously] why cautiously?

This word has been removed in the revised manuscript.

L478: [For this direct route, following the extraction of person-relevant features from an incoming voice signal, the common recognition mechanism matches the analysed voice signal (i.e., pattern of deviant features) directly with a stored representation of a known person.] Isn't this inconsistent with your proposal above of a single average voice prototype?

Thank you for this comment. We do not think that this statement is inconsistent with the proposal of having a single average voice prototype. Upon re-reading this sentence, however, the choice of using the word “directly” (as opposed to perhaps “via prototype-based coding”) might be ambiguous and may cause confusion. We have therefore removed this word from the sentence and hope this addresses the reviewer’s comment:

P14: “For this direct route, following the extraction of person-relevant features from an incoming voice signal, the common recognition mechanism (see Section 5.2) matches the analysed voice signal with a stored representation of a known person”

If there is another issue with this sentence that is not covered by this edit, we'd be happy to make further changes to this sentence.

L547: [The multivariate nature] perhaps multi-attribute would be more appropriate?

Thank you for this suggestion – we have changed the wording accordingly.

L602: [.. As acknowledged] remove extra dot

Done

13th Apr 23

Dear Nadine,

Your manuscript titled "A model for person perception from voices." has now been seen by the same reviewers as before, whose comments appear below. In light of their advice I am delighted to say that we are happy, in principle, to publish a suitably revised version in Communications Psychology under the open access CC BY license (Creative Commons Attribution v4.0 International License).

We therefore invite you to revise your paper one last time to address a list of editorial requests. We ask that you edit your manuscript to comply with our format requirements and to maximise the accessibility and therefore the impact of your work.

To guide you in your revisions, I have attached a version of your manuscript that contains editorial comments.

EDITORIAL REQUESTS:

SUBMISSION INFORMATION:

OPEN ACCESS:

Communications Psychology is a fully open access journal. Articles are made freely accessible on publication under a [CC BY license](http://creativecommons.org/licenses/by/4.0) (Creative Commons Attribution 4.0 International License). This license allows maximum dissemination and re-use of open access materials and is preferred by many research funding bodies.

At acceptance, you will be provided with instructions for completing this CC BY license on behalf of all authors. This grants us the necessary permissions to publish your paper. Additionally, you will be asked to declare that all required third party permissions have been obtained.

* **TRANSPARENT PEER REVIEW:** Communications Psychology uses a transparent peer review system. On author request, confidential information and data can be removed from the published reviewer reports and rebuttal letters prior to publication. If you are concerned about the release of confidential data, please let us know specifically what information you would like to have removed.

Please note that we cannot incorporate redactions for any other reasons.

[link redacted]

Best wishes,

Marike

Marike Schiffer, PhD
Chief Editor
Communications Psychology

REVIEWERS' COMMENTS:

Reviewer #1 (Remarks to the Author):

The authors have adequately answered most of my comments. Two open points remain:

1. In response 1b to reviewer 3, the authors write regarding prototype models: "However, when more carefully thinking about the plausibility and when seeking the empirical basis for including prototypes in these earlier models (Lavner/Maguinness), we could not find a comprehensive account in either paper." And then the authors write that a support for prototype models are the findings that the "clearest justification we can see for invoking prototypes and calculating deviance patterns is to account for empirical observations that face and voice identities that are highly deviant from the prototype/average are perceived as distinctive and are more accurately recognised".

I'm surprised that the authors mention that there is no comprehensive account in either paper. Lavner et al., is an empirical test of a prototype model and in Maguinness et al., 2018 there is a rather large section (see page 2) on the three types of findings that support prototype processing. These are 1) more distinct voices are recognised more robustly than average counterparts 2) voice samples which emphasise speaker-specific deviations, i.e. vocal caricatures, are more readily recognised as belonging to an identity than more veridical voice samples 3) perceptual aftereffects which are consistent with prototype-based processing of voice-identity. For details and references see Maguinness et al., 2018 page 2.

2. Original comment 12: The authors mentioned that they have addressed the comment, but in the revised manuscript there is still no reference to Maguinness et al., 2018 in the sentence that list those models that include a mirrored face part.

Reviewer #2 (Remarks to the Author):

Thank you for your revision work to a manuscript that, I hope, will be very useful to the community working on the processing of complex sounds at large.